# Whole Genome Sequencing Applied in Familial Hamartomatous Polyposis Identifies Novel Structural Variations

**DOI:** 10.3390/genes13081408

**Published:** 2022-08-08

**Authors:** Revital Kariv, Dvir Dahary, Yuval Yaron, Yael Petel-Galil, Mira Malcov, Guy Rosner

**Affiliations:** 1Department of Gastroenterology, Tel Aviv Sourasky Medical Center, Tel Aviv 64239, Israel; 2Faculty of Medicine, Tel Aviv University, Tel Aviv 69978, Israel; 3Geneyx Ltd., Tel Aviv 6805018, Israel; 4IVF Unit, Tel Aviv Sourasky Medical Center, Tel Aviv 64239, Israel

**Keywords:** hamartomatous polyposis, whole-genome sequencing, structural genomic, inversion, deletion

## Abstract

Hamartomatous polyposis syndromes (HPS) are rare cancer-predisposing disorders including Juvenile polyposis (JPS), Peutz–Jeghers (PJS) and PTEN hamartomatous syndromes (PHS). Penetrant mutations in corresponding genes (SMAD4, BMPR1A, STK11, PTEN and AKT1), are usually diagnosed via a next-generation-sequencing gene panel (NGS-GP) for tailored surveillance and preimplantation testing for monogenic disorders (PGT-M). Five probands with HPS phenotype, with no genetic diagnosis per genetic workup, underwent whole-genome sequencing (WGS) that identified structural genetic alterations: two novel inversions in BMPRA1 and STK11, two BMPR1A-deletions, known as founders among Bukharan Jews, and BMPR1A microdeletion. BMPR1A inversion was validated by “junction fragment” amplification and direct testing. PGT-M was performed via multiplex-PCR and enabled successful birth of a non-carrier baby. WGS may be considered for HPS patients with no NGS-GP findings to exclude structural alterations.

## 1. Introduction

Hamartomatous polyposis syndromes (HPS) are severe infrequent cancer-predisposing disorders with estimated incidence of 1/30,000–1/200,000, presenting with multiple hamartomatous polyps (HP) and adenomas throughout the gastrointestinal tract [1]. Syndromes and corresponding genes include: SMAD4 (DPC4) and BMPR1A for juvenile polyposis (JPS), STK11 for Peutz–Jeghers (PJS) and PTEN and AKT1 for PTEN hamartomatous syndromes (PHS) [1,2]. Clinical and histological overlap exist between various HPs [3], yet cancer susceptibilities differ. Familial mutation identification enables segregation and appropriate surveillance, as well as preimplantation genetic testing for monogenic disorders (PGT-M) [4]. Currently, next-generation-sequencing-based gene panels (NGS-GP) and whole exome sequencing (WES) are used for diagnosis; however, they yield margins of 50–70% [5].

WES expands the gene repertoire for single and short nucleotide variants (SNV), but cannot accurately and fully detect structural variants (SV) and copy number variations (CNV) [4,6]. Whole-genome sequencing (WGS) has recently emerged as an excellent diagnostic tool for infrequent diseases, enabling identification of genomic mutations and SVs or CNVs such as large deletions, duplications and inversions. Indeed, large genomic deletions in SMAD4 and BMPR1A can explain JPS [7,8,9,10,11]. WGS is not sensitive to an enrichment kit; therefore, it results in high-quality variant calling [6]. The vast amount of raw sequencing data, exhaustive variant calling, cost and lack of reliable analytic tools are barriers for WGS use. We present WGS analysis results identifying unique SV in a small HPS cohort.

## 2. Patients and Methods

Five non-related probands with HPS phenotype, without identified pathogenic mutations, were offered WGS analysis. First-degree family members were used for validation and segregation analyses.

### 2.1. Methods

#### Local and National Ethics Committees Approved the Study

Among fifty-two individuals from 34 families with JPS, PJS or PHS diagnosis according to NCCN criteria, (https://www.nccn.org/professionals/physician_gls/pdf/genetics_colon.pdf, accessed on 1 November 2021) who were identified at our center 2004–2017, we offered five probands WGS analysis, as routine workup at that time did not detect any pathogenic mutation. Suspected pathogenic variants including structural variants, found by the analysis platform, were validated at certified genetic labs.

We conducted whole-genome sequencing (WGS) from germline DNA on the Illumina sequencing platform to obtain a mean coverage of 30X at Centogene AG, Rostock 18055, Germany. Data analysis was performed by using the Geneyx Analysis platform, formerly branded as the TGex software [5].

The Analysis platform was used for extraction of raw sequencing data from the sequencing provider, followed by primary and secondary pipelines to generate VCF files with SNVs, SVs, CNVs and repeats. VCF files went through a comprehensive annotation pipeline for analysis, and all the resulting annotated variants were displayed in an interactive user interface for analysis interpretation and selection of plausible candidates in the context of the clinical conditions presented by each patient. The annotation pipeline, as well as the annotation database and architecture of a variant interpretation of the platform, is mentioned in detail elsewhere [5]. Briefly, the analysis is based on the standard steps of basic variant annotation, allele frequency databases and variant damage prediction, and offers phenotype-driven interpretation that relies on a comprehensive knowledge base and annotation of structural variants.

## 3. Variant Validation

The BMPR1A inversion (Appendix A) was validated by designing two sets of PCR primers, amplifying both edges of the inversion breakpoints—one set in the normal allele as and the other set in the mutated one, as demonstrated in Figure 1.

Segregation analysis (Figure 2) for BMPR1A-linked haplotypes was based on 17 informative polymorphic markers, 5 of which are localized within the inverted region. The other 12 are flanking the BMPR1A gene.

Segregation analysis for STK11-linked haplotypes was based on 20 informative polymorphic markers flanking the STK11 gene. Haplotype characterization confirmed the linkage of the suspected inversion with clinical familial segregation, and it allowed the use of a multi-loci detection methodology for preimplantation genetic testing for monogenic gene (PGT-M) for the BMPR1A inversion.

## 4. Preimplantation Genetic Testing for Monogenic Disorders (PGT-M)

For PGT-M, the proband’s spouse underwent hormonal stimulation followed by aspiration of 16 mature oocytes. Single blastomeres were biopsied from 12 embryos, 3 days following fertilization, and subjected to single-cell multiplex PCR as previously described [12].

For the genetic analysis, paralleled amplifications of two regions bordering the inversion break points, together with eight flanking informative polymorphic markers, were performed. Eight embryos clearly demonstrated the paternal normal allele and were found suitable for transfer. The other four embryos presented the inverted BMPR1A allele inherited from the affected father, and therefore were not suitable for transfer. Due to obstetric considerations, healthy embryos were frozen and a transfer of a thawed single healthy embryo, a month later, resulted in a successful pregnancy and birth of a healthy, unaffected baby.

## 5. Results

**Proband A** A 30–35Y Ashkenazi Jewish male was diagnosed with dozens of colorectal and gastric hamartomatous and adenomatous polyps. The proband’s mother and brother had each >10 colorectal and gastric polyps. The maternal grandfather had colorectal cancer (CRC) at age 35. Genetic workup, found to be normal, included: SMAD4, BMPR1A Sanger sequencing (SS), multiple ligation probe assay (MLPA) and WES. WGS revealed novel unique intragenic BMPR1A inversion (NC_000010.10:g.87852798_88575769inv), supporting the diagnosis of JPS. Molecular validation was performed using the “junction fragment” amplification (Figure 1). Familial segregation via haplotyping and direct testing of the inversion revealed the proband’s mother and brother, but not the maternal aunt, were affected (Figure 2). PGT-M by multiplex PCR was performed with a successful pregnancy and birth of a healthy non-carrier baby.

**Proband B** A 45–50Y Jewish female of Turkish–Syrian origin presented with Peutz–Jeghers-type colorectal, gastric and duodenal HP and adenomas from age 28. At 38Y, papillary thyroid carcinoma was diagnosed. Physical examination showed internal lower lip hyperpigmentation. Family history included prostate cancer in the proband’s father at 65Y. Genetic workup included STK11 SS, MLPA and NGS-GP, which were all normal. WGS revealed a unique novel intragenic inversion in the STK11 gene (NC_000019.9:g.1206071_1274737inv), supporting PJS. Detailed haplotype segregation in the extended family was performed (father, mother, two brothers and a daughter, Figure 2), while only the 17Y daughter presented a relevant phenotype (lip hyper-pigmentation). Analysis demonstrated the patient’s paternal allele in the daughter. This hint for de novo intragenic STK11 gene inversion may have occurred on the paternal allele during spermatogenesis or in an early stage of the patient’s embryonic development.

**Proband C** A 50–55Y Bukharan Jewish female was diagnosed in her twenties with dozens of colorectal HPs and adenomas and few gastric and small-bowel HPs. At 53Y, she was diagnosed with breast cancer. Both her sons, identical twins, presented with rectal bleeding at age 18 with >10 colorectal HP. Sister and nephew presented similarly. Her father had 10 colonic HPs, and her paternal grandfather had CRC at 54Y. BMPR1A and SMAD4 SS were found to be normal in the proband and her two sons. WGS analysis detected BMPR1A gene 11-exons deletion (NC_000010.10:g.88611710_88871320del), similar to previously reported deletion among Bukharan Jewish families [8], supporting JPS diagnosis. This deletion could have been detected by MLPA testing.

**Proband D** A 35–40Y Bukharan Jewish male was diagnosed at age 14 with multiple colorectal HP and adenomas. Father and paternal grandfather had CRC in their 60s. The proband’s 13-year-old daughter had >20 colorectal HPs and adenomas. BMPR1A and SMAD4 SS were found to be normal. WGS detected BMPR1A gene 11-exons deletion (the same as in unrelated case C), in the affected father and his daughter, supporting JPS.

**Proband E** A 40–45Y Ashkenazi Jewish male was diagnosed with dozens of colorectal hamartomatous and adenomatous polyps. Due to high polyp burden, he underwent subtotal colectomy with ileorectal anastomosis. His brother (47Y) had a few colorectal adenomas and HP. His father had CRC at 67Y. BMPR1A and SMAD4 sequencing was normal. WGS detected BMPR1A short indel mutation that caused a frameshift (NM_004329.2:c.1419del; p.Val474Cysfs *24), supporting JPS. This mutation could have been detected by NGS-GP.

Figure 2 and Appendix A describe pedigrees and segregation analysis and WGS findings, respectively.

## 6. Discussion

We present five families with HPS phenotype (4 JPS,1 PJS), in whom commonly practiced genetic workup was found normal.

WGS analysis detected pathogenic mutations in all probands, while segregation analysis supported their pathogenic nature. One subject had a small BMPR1A gene micro-deletion, while in all others, SV were found. These included two Bukharan-origin subjects with BMPR1A 11-exons deletion, previously reported by Liberman et al. [8]. SV in BMPR1A are uncommon, and encompass ~6% of cases [8,9]. However, we are unaware of reports on BMPR1A inversion, as well as STK11-pathogenic SV. WGS, which is rarely performed for HPS, was used with a user-friendly bioinformatics software analysis that enabled a clinical team to pinpoint the causal variants. Validation and segregation of the BMPR1A inversion enabled in vitro fertilization (IVF) with PGT-M based on direct mutation testing and haplotype analysis, resulting in a normal pregnancy and birth of an unaffected baby.

SV/CNV are expected to have a more severe clinical phenotype compared to missense mutations, due to widespread functional effects, as well as possible effects on downstream genes [9]. Indeed, the phenotype of affected subjects in our cohort showed early onset advanced polyps, as well as neoplastic gastric polyps among BMPR1A SV carriers, which are less expected in BMPR1A mutation carriers [7,8]. Similarly, the STK11 pathogenic inversion carrier presented with a severe clinical phenotype.

In conclusion, WGS seems to be an effective tool to detect SV in HPS cases, especially when NGS-GP is interpreted as normal. Novel intragenic inversions are part of the HPS SV spectrum. Future studies in larger cohorts are needed to evaluate SV contribution to HPS.

## Figures and Tables

**Figure 1 genes-13-01408-f001:**
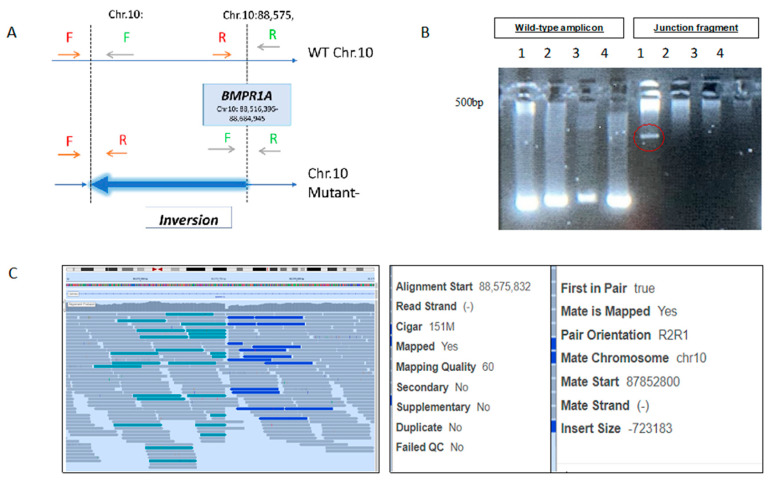
(**A**) Schematic analysis of BMPR1A inversion: In the wild-type state, chr.10 cannot be amplified by the combination of F up and R up or F down and R down primers, due to the distance between them and their orientation. The dashed lines indicate the break points of the inversion. In affected individuals, both primer pairs are in proximity and alignment that allow the amplification of the inversion junction fragment and, as a result, the detection of ~200 bp PCR products (bold arrow). (**B**) Sample 1 represents the affected proband, while the other 3 samples are of healthy controls. The normal region of the gene-wild-type amplicon is demonstrated in all DNA samples, while it is amplified in the proband. However, the junction fragment that represents the mutant allele was amplified only in the proband that carries the BMPR1A inversion (enhanced by a red circle). (**C**) IGV snapshot of the BMPR1A breakpoint of the identified inversion and an example of a read pair in which both reads have the highest mapping quality, yet the first is at position 88.5 M and the second is mapped to 87.8 M on the same chromosome and on the opposite strand (both mapped minus, while usually strands of paired reads should be opposite).

**Figure 2 genes-13-01408-f002:**
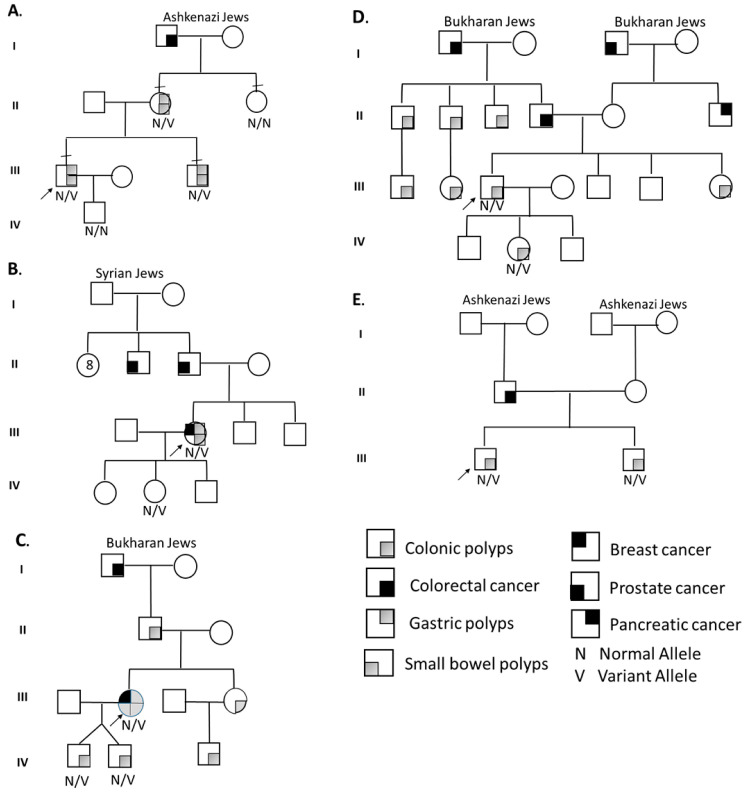
Pedigrees and segregation analysis results of the study families. ((**A**–**E**) correspond to probands order).

## Data Availability

Data available on request due to restrictions e.g., privacy or ethical. The data presented in this study are available on request from the corresponding author. The data are not publicly available due to privacy and confidentiality.

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
