# Peer review of "Whole Genome Sequencing Applied in Familial Hamartomatous Polyposis Identifies Novel Structural Variations"

_genes, 2022, doi:10.3390/genes13081408_

Round 1

Reviewer 1 Report

This paper reports a very high rate of identification of causative genetic alterations on WGS in a group of patients with the phenotype of rare hamartomatous polyposis, but with no genetic cause identified by NGS.

This is of obvious benefit to the individuals concerned and to their families, and clearly demonstrates the need for WGS under these circumstances. As such it is a useful addition to the literature.

I do not think any changes are needed.

Author Response

We would like to thank the reviewer for these comments and agree that this approach may benefit these families. 

Reviewer 2 Report

The report by Kariv and Rosner shows that whole genome sequencing was used to detect genetic alteration in persons affected by Familial Hamartomatous Polyposis that whose typical genetic alterations were not detected by conventional methods. The report is interesting but can be improved by addressing the following concerns:

- Figure 2 is referenced before figure 1 and is shown before figure 1.

- It is known that WES can sequence intragenic regions beyond exons. Given that WGS was performed because WES was unable to detect genomic alteration, the evidence from WES and from WGS for at least one case is needed.

- A figure showing how the WGS reads determined inversion in BMPR1A is needed. The figure 1A shown is only a schematic summarizing the findings, but not the actual findings.

- The supplementary material is illegible. I suggest to include an index at the beggining (even if it is short) and then each of the evidence referred and needed for probands.

Author Response

We thank reviewer #2 for the detailed review and we answer the reviewer point by point as below.

The report by Kariv and Rosner shows that whole genome sequencing was used to detect genetic alteration in persons affected by Familial Hamartomatous Polyposis that whose typical genetic alterations were not detected by conventional methods. The report is interesting but can be improved by addressing the following concerns:

- Figure 2 is referenced before figure 1 and is shown before figure 1. THANK YOU FOR THIS COMMENTS WE HAVE CHANGED THE ORDER OF THE FIGURES TO MATCH QUOTATION AT THE TEXT. FIGURE LEGEND WAS UPDATED AS BELOW.

- It is known that WES can sequence intragenic regions beyond exons. Given that WGS was performed because WES was unable to detect genomic alteration, the evidence from WES and from WGS for at least one case is needed. WE THANK THE REVIEWER FOR THIS INPUT. ACTUALLY THE INITIAL GENETIC WORKUP INCLUDED EITHER SANGER SEQUENCING OF THE RELEVANT GENE AND/OR NGS PANELS. NONE OF THE FAMILIES UNDERWENT WES, AS CASES WITH OBVIOUS PHENOTYPE  ARE OFTEN OFFERED NGS PANELS THAT INCLUDE THE RELEVANT GENES. PROBANDS THAT UNDERWENT NGS PANEL HAD COMPLETELY NORMAL RESULTS. IT IS MENTIONED IN THE TEXT FOR EACH PROBAND WHAT TYPE OF GENETIC TESTS PRECEDED THE WGS.

- A figure showing how the WGS reads determined inversion in BMPR1A is needed. The figure 1A shown is only a schematic summarizing the findings, but not the actual findings. WE ACCEPT THE REVIWER'S SUGGESTION AND HAVE ADDED THE REQUIRED INFORMATION TO FIGURE 1 AS "PANEL C". SINCE THIS IS A VERY SPECIFIC INFORMATION WE SUGGEST THE EDITOR'S CONSIDERATION IF THIS INFORMATION SHOULD BE IN THE MAIN TEXT OR IN SUPPLEMENTARY DATA. WE WILL SEND SEPARATE LY IN ADDITION TO THE MAIN DOCUMENT A PPT FILE OF FIGURE 1 FOR POTENTIAL EDITS.

  • The supplementary material is illegible. I suggest to include an index at the beggining (even if it is short) and then each of the evidence referred and needed for probands. WE THANK THE REVIEWER. ACTUALLY THE SUPPLEMENTARY DATA WAS NOT MEANT TO BE PUBLISHED IN TOTAL SO WE APOLOGIZE FOR THE MISUNDERSTANDING.   THIS PREVIOUSLY UPLOADED SUPPLEMENTARY DATA INCLUDED MUTATION TABLE AND COMPLETE MUTATION REFERENCE AND IN ADDITION THE CLINICAL INFORMED CONSENTS THAT ARE TYPICALLY BEING ASKED FOR IN SUCH STUDIES, OBVIOUSLY THISE CANNOT BE PUBLISHED AND JUST SERVE AS AN EVIDENCE FOR PATIENT CONSENT. WE NOW EXCLUDED THE CONSENTS IF NOT ASKED FOR AND INCLUDED ONLY THE TABLE OF GENETIC FINDINGS AS "SUPPLEMENTARY TABLE 1" AT THE END OF THE MAIN FILE. IF REVIEWERS OR EDITORS ASK FOR A CHANGE OR EDITS PLEASE INFORM US. 
